# Neuroendocrine Neoplasms of the Female Genitourinary Tract: A Comprehensive Overview

**DOI:** 10.3390/cancers14133218

**Published:** 2022-06-30

**Authors:** Mayur Virarkar, Sai Swarupa Vulasala, Dheeraj Gopireddy, Ajaykumar C. Morani, Taher Daoud, Rebecca Waters, Priya Bhosale

**Affiliations:** 1Department of Diagnostic Radiology, University of Florida College of Medicine, 655 West 8th Street, C90, 2nd Floor, Clinical Center, Jacksonville, FL 32209, USA; mayur.virarkar@jax.ufl.edu (M.V.); dheeraj.gopireddy@jax.ufl.edu (D.G.); 2Department of Diagnostic Radiology, The University of Texas MD Anderson Cancer Center, 1515 Holcombe Blvd., Houston, TX 77030, USA; amorani@mdanderson.org (A.C.M.); tedaoud@mdanderson.org (T.D.); rwaters@mdanderson.org (R.W.); priya.bhosale@mdanderson.org (P.B.)

**Keywords:** neuroendocrine tumors, female neuroendocrine neoplasms, genitourinary NENs, cancer staging, imaging of neuroendocrine tumors

## Abstract

**Simple Summary:**

Primary neuroendocrine neoplasms (NENs) are a rare, heterogeneous group of tumors that include well-differentiated neuroendocrine tumors, poorly differentiated neuroendocrine carcinoma, and paraganglioma. NENs in the urinary tract are observed in <0.05% of individuals, in whom the bladder is the most common site. In this review, we described the epidemiology, pathogenesis, imaging, staging, and management of the genitourinary NENs.

**Abstract:**

Primary neuroendocrine neoplasms are a rare heterogeneous group of tumors that include well-differentiated neuroendocrine tumors, poorly differentiated neuroendocrine carcinoma, and paraganglioma. An extensive literature search was used to compile the data regarding epidemiology, pathogenesis, imaging features, and management of the urinary system NENs. We also included the updated staging of the NENs at various locations of the urinary system.

## 1. Introduction

Neuroendocrine neoplasms (NENs) are a diverse group of tumors arising from neuroendocrine cells and share similar features throughout the body. NENs in the urinary tract are observed in <0.05% of individuals, in whom the bladder is the most common site (Figure 1) [1]. The free radical and genetic damage secondary to smoking are the most important risk factors for urothelial carcinoma and urinary tract NENs [2]. This manuscript discusses the epidemiology, pathogenesis, imaging, and management of NENs in the kidney, ureter, bladder, and urethra.

## 2. Renal NENs

### 2.1. Epidemiology, Presentation, and Pathogenesis

Primary renal NENs are rare neoplasms with around 100 cases reported in the literature, as per our knowledge (Figure 2) [3]. The 2016 WHO classification of renal tumors classified renal neuroendocrine neoplasms into well-differentiated NET, small-cell NEC, and large-cell NEC (Table 1) [4]. Renal carcinoid tumors, also known as well-differentiated NET of the kidneys, are rare, slow-growing neoplasms. They were first described by Resnick et al. in 1966 [5,6]. In a study by Hansel et al., the median age of presentation for renal NENs is 52 years, and the tumor equally affected male and female patients [7]. Primary renal carcinoid is the second most common genitourinary carcinoid tumor after testicular/ovarian carcinoids [8]. It is asymptomatic and incidentally detected in 25–30% of patients [6,9]. Typical symptoms of the carcinoid syndrome, including flushing, edema, and diarrhea, are uncommon and are seen in <10–15% of patients [5,10,11,12]. The incidence of small-cell renal NEC is rare; approximately 50 cases have been reported in the literature [13,14,15,16,17]. Most of the patients present at the age of 59 years with non-specific symptoms, such as hematuria, weight loss, and abdominal pain. There is no gender predilection, and the right kidney is more commonly involved than the left, with a ratio of 1.5:1 [13]. Evaluation with biochemical tests such as urinary–HIAA and serum chromogranin A and imaging is recommended to assess suspected NENs [18]. Death is seen in around 75% of patients within a year of diagnosis. Primary renal large cell NEC is an extremely rare malignancy; fewer than 10 cases have been reported in the literature [14,19,20,21,22,23,24]. Renal paragangliomas are rare, and to date, very few cases have been reported in the literature [25].

### 2.2. Pathogenesis

There are many hypotheses explaining the cellular origins of neuroendocrine neoplasms in the kidneys: tumors (i) arise from renal stem cells that progress towards neuroendocrine lineage [18] or (ii) arise from entrapped neural crest cells in the kidney during embryogenesis [33] or (iii) develop alongside congenital abnormalities of the kidney [18] or (iv) are secondary to the metaplasia of pyelocaliceal urothelium resulting from chronic inflammation [26]. Renal NENs arises most commonly in individuals with congenital or acquired renal anomalies, such as horseshoe kidney (26% of renal NENs), polycystic disease, and mature cystic teratoma (15% of renal NENs) [34,35,36,37,38]. Krishnan et al. speculated that renal NENs arises from the already existing hyperplasia of neuroendocrine cells in the metastatic or teratomatous epithelium of the horseshoe kidney [34]. Primary renal small-cell NEC is a poorly differentiated tumor with a poor prognosis that is similar to small-cell NEC at other sites. Unlike renal parenchymal small-cell NEC, the ones arising from the renal pelvis are associated with urothelial or squamous cell carcinoma [17,39,40]. The origin of renal small-cell NEC is different from that of clear cell renal cell carcinoma. The pattern of chromosomal aberration can explain this. Gain of multiple chromosomes, loss of *p53*, and amplification of *MYC* are seen in small-cell renal NEC, whereas loss of the short arm of chromosome 3 (3p) is observed in clear cell renal cell carcinoma [28]. However, El-Naggar reported a loss of heterozygosity in chromosome 3p21 and suggested that this event is preliminary to all the renal neoplasms [41]. Renal paragangliomas are usually benign, and around 10% of patients encounter malignant transformation [29]. Classic cancer predisposition syndromes, including von Hippel–Lindau, multiple endocrine neoplasia type-2, and neurofibromatosis type 1 disease are associated with paragangliomas. Metastases are rare in *VHL* and *MEN2* associated paragangliomas, whereas they may occur in 12% of *NF-1* associated diseases [42]. The risk of metastases is high in succinate dehydrogenase subunits B (25–50%) and A (12%) [43].

### 2.3. Imaging

Ultrasound of renal NENs shows an iso-to hyperechoic mass with a peripheral anechoic or hypoechoic rim [11,12,44]. Computed tomography demonstrates a well-circumscribed solid, hypodense tumor with minimal enhancement in contrast-enhanced venous phase CT that corresponds to hypo- or avascular lesions on angiography (Figure 3, Figure 4 and Figure 5) [18,45]. Heterogeneity and calcifications on CT can be seen in 60% and 26% of cases [9,33]. Around 75% of patients present with tumors > 4 cm, and >50% of patients demonstrate extrarenal invasion (perinephric, renal vein, or renal sinus fat) or distant spread of tumor [12,46]. However, differentiation is challenging, solely based on CT or MR imaging findings among renal NENs and renal cell carcinoma (Table 2). In addition, smaller carcinoids are difficult to visualize on conventional imaging. Octreotide scintigraphy complements CT or MRI and aids in diagnostic evaluation, staging, and follow-up of tumors and metastases with high sensitivity (85%) [13,47]. However, the major limitation is the uptake of tracer material by the normal renal parenchyma, obscuring the suspicious lesion.

Renal paragangliomas are more common in the kidney’s upper pole, and all patients shall have a biochemical evaluation before surgical resection [49]. Patients with hereditary syndrome-associated paragangliomas are more likely to develop multifocal primary tumors with varying degrees of metastases [50]. Paragangliomas are well-defined tumors with hemorrhagic and necrotic areas on CT, and calcification in 15% of cases [51]. On contrast administration, they exhibit intense homogeneous enhancement secondary to the tumor hypervascularity [51]. Since the MR resolution of tumors is better than that of CT, MR imaging has higher sensitivity in detecting liver metastases and blood vessel invasion [52]. In MR imaging, paragangliomas demonstrate low to intermediate signal intensity on T1-weighted, signal voids on T1-weighted spin-echo, and high signal intensity on T2-weighted imaging sequences [51]. Functional imaging, iodine 123 metaiodobenzylguanidine (I^123^-MIBG), is widely used to identify metastases and differentiate functional from nonfunctional paragangliomas.

I^123^-MIBG has high specificity (95–100%) but a low detection rate; the sensitivity ranges from 56 to 72% [53,54,55,56]. The application of PET/CT with 68Ga-DOTA peptides and 18F-fluorodeoxyglucose has increased recently. PET/CT imaging with 68Ga-labeled somatostatin analogs such as DOTATOC, DOTATATE, and DOTANOC is being employed, and it targets the over-expressive somatostatin receptor subtype 2 in the paragangliomas [57]. DOTATATE PET/CT is the better imaging modality in patients with well-differentiated renal NENs. 18F-FDG PET/CT is superior to the 68Ga-based PET/CT, 123I-MIBG, CT, or MRI for high-grade NEC and detects the metastatic or recurrent tumors. It exhibits increased 18F-FDG uptake in SDH and VHL-related tumors and decreased uptake in MEN-2-related tumors. In addition, fluorine-18-labeled dihydroxyphenylalanine, a new tracer, was found to acquire images within hours; compare that to a day for an MIBG scan. The main limitation of FDG-PET is that its specificity is lower than that of 123I-MIBG and 111-In Octreoscan [58,59].

The diagnosis of NENs is based on histopathological and immunohistochemical characteristics of the tumor (Figure 6). Grossly, the renal NENs is solitary with well-delineated borders from the surrounding normal parenchyma and demonstrates solid (74% of cases) and cystic (49% of cases) surfaces, alongside focal calcification (30% of cases) and the occasional pseudo-capsule [47]. Usually, as the retroperitoneal space is highly distensible, the patients manifest with large tumor masses at diagnosis [33]. In their study, Hansel et al. described the average tumor size being from 2.6 to 17 cm. They reported that around 62% of patients demonstrated metastasis in regional lymph nodes or distant organs [3,7]. Hence, it can be said that the metastases are directly related to the tumor size in the renal NENs [3,60].

### 2.4. Prognosis and Management

Renal NENs have higher survival rates than poorly differentiated renal NENs [3]. Age > 40 years, tumor size > 4 cm, extrarenal tumor invasion, purely solid surface, and >1 mitoses per HPF are the poor prognostic factors in patients with renal NENs [9,10,61]. In the presence of horseshoe kidneys, the carcinoids manifest indolent behavior [12,33,62]. Zhenglin et al. reported regional or distant metastases in 83% and a median overall survival of only 27 months in patients with renal NEC [3]. Patients with renal paraganglioma and metastases at diagnosis have 5-year survival rates ranging from 50 to 70% [30].

Surgery is the mainstay of treatment in renal NENs. The staging of renal NENs is described in Table 3 and Table 4. The standard treatment for localized primary renal NENs is nephrectomy with a lymph node dissection (Figure 7) [18]. As the surgery could be a prognostic factor, radical or partial nephrectomy is the treatment choice for localized or metastatic renal NENs [33]. Resection of the entire visible tumor is recommended in localized renal NEC. Rosenberg et al. described that, as carcinoids tend to be benign, partial nephrectomy or cryoablation of the tumor shall be the standard treatment; therefore, functional nephrons can be spared [63]. Chemotherapy and radiation have no proven benefit in managing NENs in general [13]. When indicated, open surgical resection shall be considered in the case of primary paraganglioma. Cytotoxic chemotherapy is recommended as the first-line therapy in patients with a high tumor burden (extensive metastases), symptoms, or rapid progression. Clinically used chemotherapeutic medications include CVD (cyclophosphamide/vincristine/dacarbazine) or temozolomide-based regimen. ^I131^ MIBG therapy can be considered in patients requiring systemic treatment and who have MIBG-avid disease [30].

## 3. Urinary Bladder NENs

### 3.1. Epidemiology, Presentation, and Pathogenesis

Urinary bladder NENs are classified into well-differentiated NENs, small-cell NENs, large-cell NENs, and paragangliomas (Table 5) [28]. They comprise <1% of the urinary bladder tumors and are more common in males with a male-to-female ratio of 3.3:1.0 [64,65]. A small-cell variant is the common NENs of the urinary bladder, with an estimated 500 cases per year [64]. It constitutes 0.3–0.7% of primary bladder malignancies [66]. Small-cell NEC of the urinary bladder was first described by Cramer et al. in 1981 [66]. Note that 88% of small-cell NEC patients presented with hematuria in a study by Cheng et al. [65]. Around 60% of patients are diagnosed with metastatic disease at diagnosis [67]. Cigarette smoking is a risk factor in 50–70% of small-cell bladder NECs [68]. Additionally, small-cell NEC is predominant in the sixth to seventh decades of life. In contrast, well-differentiated NENs of the urinary bladder are rare and are usually seen in middle-aged individuals; there is a slight predominance in males [69,70]. Large-cell NEC constitutes <0.5% of all the bladder urothelial carcinomas and is usually reported in male (77.1% vs. 23%), elderly patients (>60 years) [24,71,72]. In the urinary tract, the bladder is the most common site for the origin of large-cell NEC [73,74]. The urinary bladder (79.2%), followed by urethra (13%), pelvis (5%), and ureter (3%) are the most common sites of paraganglioma in the genitourinary tract [75,76]. The first case of urinary bladder paraganglioma was reported in 1953 by Zimmerman et al. [77]. Bladder paragangliomas account for 0.05–0.1% of all bladder tumors and 6% of paragangliomas [78,79]. Females are three times more commonly affected than males [80]. Functional paragangliomas secrete catecholamines resulting in hypertension, diaphoresis, headache, palpitations, and post-micturition syncope [81]. The characteristic triad of paroxysmal or sustained hypertension, gross hematuria, and micturition syncope can be observed in >75% of patients [46,82].

Well-differentiated NENs are derived from the neuroendocrine cells in the normal urothelial and reactive urothelial basement membrane. There are three proposed theories to explain the cells of origin of small-cell urinary bladder NENs: (i) From undifferentiated and multipotent cells or stem cells in the urothelium. This theory supports the frequent co-existence of small-cell NEC and urothelial carcinoma. (ii) From neuroendocrine cells of the regular or metastatic urothelium and (iii) from undefined submucosal neuroendocrine cells [83]. While 10–30% of patients present with a pure form of small-cell NEC, around 70–90% of patients, small-cell carcinoma coexists with adenocarcinoma, squamous cell carcinoma, or urothelial carcinoma [65,84]. Large-cell NEC of the urinary bladder can occur as a pure variant or associated with squamous cell carcinoma, carcinosarcoma, primary adenocarcinoma of the bladder, or urothelial carcinoma [85]. Pheochromocytomas at the extra-adrenal sites are termed paragangliomas. They arise from the sympathetic chain in the detrusor muscle and may occur anywhere in the urinary bladder [46,82]. Although most of the cases are sporadic, a few cases of bladder paraganglioma are associated with hereditary syndromes, such as neurofibromatosis, Sturge–Weber syndrome, von Hippel–Lindau syndrome, tuberous sclerosis, and the Carney triad (pulmonary chondromas, paragangliomas, and gastrointestinal stromal tumors) [86].

### 3.2. Imaging

Well-differentiated NENs of the urinary bladder appear as polyps or submucosal nodules with occasional inflammatory changes in cystoscopy (Figure 8) [69]. Most of the tumors are small, ranging from 2 to 10 mm, and involve neck and trigone regions of the bladder [69]. The CT and MR imaging of a small-cell NEC of the urinary bladder shows a broad-based polypoid mass with or without necrotic and cystic areas (Figure 9 and Figure 10). The mass displays patchy enhancement with contrast administration and exhibits invasion of the entire thickness of the bladder wall at the time of diagnosis. Bladder wall invasion is typical in tumors ranging from 3 to 8 cm in size. Kim et al. reported that the small-cell NEC of the urinary bladder is hypointense on both T1 and T2-weighted MR imaging due to the high tumor cellularity [87]. The aggressiveness of small-cell NEC of the urinary bladder is reflected in its adjacent structural invasion of the vagina, uterus, ureters, peritoneum, and abdominal muscles. Lymph node involvement can be noticed in 66% of the patients with distant metastases in the lung, liver, and bone [86,87].

Paraganglioma appears as a sub-mucosal homogeneous or heterogeneous hyper-enhancing lobulated mass on contrast-enhanced CT. A ring-like peripheral calcification is highly suggestive of a paraganglioma [88]. MR imaging is superior to CT and demonstrates low-signal intensity on T1 and moderately high signal intensity on T2-weighted imaging sequences [86]. Multiple signal voids with hyperintense foci can be seen on MR imaging, resulting in a “salt-and-pepper appearance” [78]. The enlarged feeding arteries and intense tumor blush are observed on angiography. Iodine-131 metaiodobenzylguanidine (MIBG) scan has higher specificity (96%) than CT or MRI to detect paraganglioma and aids in evaluating metastatic disease [86]. At the same time, MR imaging (88%) is more sensitive than CT (64%) or iodine-131 MIBG (64%) when identifying the extra-adrenal paragangliomas [89]. Metastatic lymph nodes are visualized by PET scan with 6-[18F] fluorodopamine scan, which is superior to iodine-131 MIBG [90]. Most of the tumors are benign, whereas 10–15% of tumors exhibit malignant behavior [91].

Immunohistochemistry reveals positivity for synaptophysin, chromogranin A, and cytokeratin AE1–AE3 [92]. Diagnosis of small-cell NEC of the urinary bladder is based on WHO diagnostic criteria for small-cell lung carcinoma, which include morphological characteristics (Table 5) [93]. Small-cell NEC of the urinary bladder stain positive for chromogranin, CD57 CD56, synaptophysin, TTF-1, neuron-specific enolase, CAM5.2, keratin7, and the epithelial membrane antigen GATA3 in immunohistochemical analysis. A dot-like reactivity to CAM5.2 can also be noticed. Synaptophysin (64.3%) and CD56 (71%) have higher sensitivity than chromogranin A (29%) in the tumor differentiation [94]. Up to 33–39% of small-cell NECs of urinary bladder demonstrate positive staining for TTF1 [95,96]. The paraganglioma appears as a well-circumscribed nodule on gross examination and shows cells arranged in the Zellballen pattern on microscopic examination (Table 5) [97]. Immunohistochemistry of the tumor reveals positive synaptophysin and chromogranin staining in chief cells, and positive S-100 and negative cytokeratin staining in sustentacular cells. Negative cytokeratin differentiates paragangliomas from urothelial and carcinoid tumors, which usually stain positive for cytokeratin [78]. Paragangliomas with succinate dehydrogenase B mutations are likely to exhibit malignant behavior [28].

### 3.3. Prognosis and Management

The pure form of well-differentiated NENs of the urinary bladder is associated with good outcomes. Most of the tumors are so small that they can be wholly resected during a biopsy. Nonetheless, 25% of patients exhibit distant or regional lymph nodal metastases [82]. The stage is the most important prognostic factor in small-cell NEC. Five-year survival rates range from 8 to 16% in low–high stage disease (Figure 11) [65,92,98]. Patients with large-cell NEC present at late stages and have a median survival of 20–23 months [68]. The presence of features such as hypertension micturition attacks, presentation at a younger age, and invasion through the bladder wall, indicates a high potential for malignant progression [82]. The differentiation solely based on gross or histological characteristics is challenging; however, the deep tissue invasion, lymph node involvement, and distant metastases indicate the malignant activity of the tumor [97]. The staging of the urinary bladder malignancies is described in Table 6 and Figure 11. Chemotherapy with/without radiation is the preferred treatment, alongside surgery, in small-cell NEC of the urinary bladder. The standard treatment of bladder paragangliomas is surgical resection with appropriate pre-and post-operative adrenergic blockade to prevent a hypertensive crisis (Figure 12). The tumor can be contracted through transurethral or open/laparoscopic radical or partial cystectomy in conjunction with lymph node dissection [78,99]. Adjuvant radiation therapy can provide better survival for individuals enduring malignant tumors [79].

## 4. Ureteral NENs

### 4.1. Epidemiology, Presentation, and Pathogenesis

Primary ureteral NENs are rare, and no more than 50 cases of small-cell ureteral NEC have been reported in the literature [2,102]. Small-cell NEC of the ureter is usually observed in elderly patients, and around 15 cases have been observed in females so far [2,102,103]. As per our knowledge, less than 10 cases of large-cell NEC of the ureter [73,74] have been reported in the literature. Neuroendocrine cells are not usually found in the urinary tract. The pathogenesis of ureteral NENs is still disputed and hypothesized to be similar to that of bladder NENs [104,105]. Most of the NENs arise from the lower ureteral segment. Collision tumors are rare, in which tumors of two or more types coexist in the same organ and do not intermix. Approximately 50% of ureteral small-cell NECs account for collision tumors [106,107]. Around 62% of patients demonstrate small-cell NEC associated with urothelial carcinoma [108].

Clinical presentation is varied, ranging from asymptomatic to complete ureteral obstruction resulting in hydronephrosis [109]. In contrast to bladder masses obstructing bilateral ureters, ureteral NENs is unilateral and does not usually cause oligo-anuria in bi-nephric individuals. Gross hematuria and flank pain are the most common symptoms of small-cell ureteral NEC. Endoscopy provides the biopsy specimen, which aids in diagnosing the mass by revealing malignant neuroendocrine cells [74,110]. A few may also present with paraneoplastic syndromes, which indicate advanced or extensive disease [102]. Ouzzane et al. reported that 72% of patients presented with pT3 or pT4 stages, and 54% developed metastases within 13 months [108].

### 4.2. Imaging

Screening of ureteral NENs is challenging due to non-specific symptoms, such as ureteral obstruction or hematuria. As the most common presentation includes hematuria and hydronephrosis, ultrasonography (USG) is the initial investigation of choice to exclude urinary tract stones. USG, although it detects ureteral distention, is insensitive to ureteral tumors. Retrograde urography can detect the luminal mass indirectly by showing a filling defect in the ureteral lumen [109]. Finally, grossly, ureteral NENs are solid and sessile; they have gray to white cut surfaces, firm consistency, and peritumoral mural thickening (Figure 13) [74,110]. Microscopically, the tumor has a desmoplastic stromal reaction, a high ki-67 index (>50–60%), necrosis, adjacent structural invasion, and lymphovascular invasion, in addition to specific cellular characteristics described in the other sections (Figure 13) [74,110,111,112].

Histologic examination of small-cell NEC of the ureter may reveal small palisading cells with scant to moderate cytoplasm, high nuclear-cytoplasmic ratios, finely speckled chromatin, and high mitotic counts (Table 7). The specimen stains positive for CAM5.2, chromogranin A, synaptophysin, neuron-specific enolase, pan-cytokeratin (AE1/AE3), and CD56 (Figure 13). Around 44% of extrapulmonary small-cell cancers stain positive for thyroid-transcription factor-1 [113,114]. Although ureteral metastases are rare, it is prudent to exclude metastatic small-cell NEC to the upper urinary tract during the diagnosis [103,115,116].

The diagnostic criteria for large-cell NEC of the ureter are based on the pulmonary counterpart: (i) Morphological characteristics of a neuroendocrine tumor (palisading, organoid nesting, trabeculae, or rosettes). (ii) High mitotic activity (>11/HPF with a median of 70/HPF). (iii) Large necrotic zone. (iv) Microscopic cellular characteristics of a non-small-cell carcinoma (large cells with low nuclear-cytoplasmic ratio, vesicular/fine/coarse chromatin, and frequent nucleoli). (v) Positive immunohistochemistry for ≥1 neuroendocrine marker (except neuron-specific enolase) and neuroendocrine granules (100–200 nm membrane-bound cytoplasmic granules) [74]. The differential diagnosis of large-cell ureteral NEC and its distinguishing features are described in Table 8.

### 4.3. Prognosis and Management

Ureteral NENs are aggressive, among which small-cell NEC has a rapid progression and dismal prognosis with a median overall survival of 17 months [113,117]. Most of the patients present with locally advanced disease, and 20% have had lymph node involvement at diagnosis [109]. The small-cell ureteral NEC’s 1- and 3-year overall survival rates are 52% and 30%, respectively [106]. The pathologic stage is the most important prognostic factor. The median survival times of patients with stages pT1–pT2 and pT3–pT4 are around 31 and 8 months, respectively [113]. Zhong et al. reported regional recurrences and distant metastasis in 9.4% and 25% of patients with early-stage disease, respectively, during a short follow-up [106]. Vimentin positive small-cell ureteral NEC had a poor prognosis in a study by Chuang et al.; however, further studies are required to determine the efficacy of vimentin in predicting the prognosis [1].

There are no well-established strategies for treating small-cell ureteral NEC due to limited data, and hence, decision-making depends on small-cell lung carcinoma treatment strategies. Oncologists consider a multimodality approach with surgery, adjuvant chemotherapy, and radiation (Figure 14) [118]. The staging of ureteral cancers is described in Table 9 and Figure 15. Radical resection is preferred in early-stage, small-cell ureteral NEC (Figure 14). However, if the patient has upper urinary tract urothelial carcinoma, nephroureterectomy with bladder cuff excision is recommended. The median survival ranges from 8.2 months with surgery alone to 24 months with adjuvant platinum-based chemotherapy [111,113,114]. Platinum-based chemotherapy with EP (etoposide and cisplatin) or CE (carboplatin and etoposide) has shown a response rate of 69% [119]. The irinotecan and cisplatin have also been shown to achieve the tumor reduction [111,120]. Studies show that adjuvant chemotherapy has more favorable outcomes compared to surgery alone. The median overall survival times for patients who received adjuvant chemotherapy and surgery alone were 24 and 12 months, respectively (*p* = 0.56) [108,113]. The efficacy of immunotherapy has not been studied in the case of small-cell ureteral NEC. Immunotherapies targeting EGFR, BCL-2, C-kit, CD56, and PDGFR-α might be promising approaches [103,121,122].

## 5. Urethral NENs

Squamous cell carcinoma is the most common urethral malignancy constituting 50–70% of cases, followed by transitional cell carcinoma and adenocarcinoma [124]. Neuroendocrine neoplasms of the urethra are rare. Most of the cases reported in the literature were mixed variants containing elements of the transitional cell, squamous cell, and adenocarcinoma. Such presentation implies the derivation of NENs from primitive precursor cells or metastatic epithelium [125].

Small-cell urethral NEC expresses both neural and epithelial markers. Neuron-specific enolase (89%), synaptophysin (50%), and chromogranin (50%) are the neural markers; and epithelial membrane antigen (56%) and carcinoembryonic antigen (57%) are the epithelial markers that can be observed in small-cell urethral NEC. Synaptophysin and chromogranin are highly specific, whereas the rest of the neural and epithelial marks exhibit poor specificity [126]. The differential diagnoses of urethral NEC are described in Table 10. Early clinical diagnosis, staging (Table 11 and Figure 16), and pathological diagnosis are imperative to improve the malignancy outcome. Aggressive treatment with surgery and adjuvant chemotherapy is recommended in limited small-cell NEC of the urethra. In the case of unresectable carcinoma, systemic chemotherapy might be considered. Studies show that cisplatin-based chemotherapy significantly predicts improved survival [127].

## 6. Conclusions

This study represents a comprehensive review of the primary urinary tract NENs. Among NENs, those arising in the urinary tract are rare and aggressive. They require multimodal treatment in most cases. The tumor diagnosis is challenging despite the imaging, molecular, and biochemical advancements. Continued research and more extensive studies are needed to determine the biological behavior and effective management of these rare tumors.

## Figures and Tables

**Figure 1 cancers-14-03218-f001:**
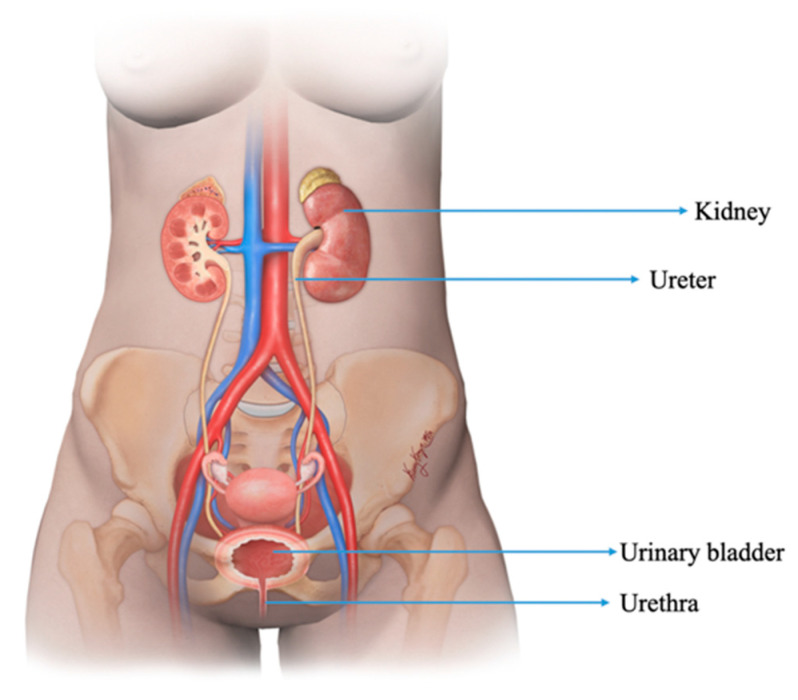
Illustration demonstrating the structures of the normal female genitourinary tract.

**Figure 2 cancers-14-03218-f002:**
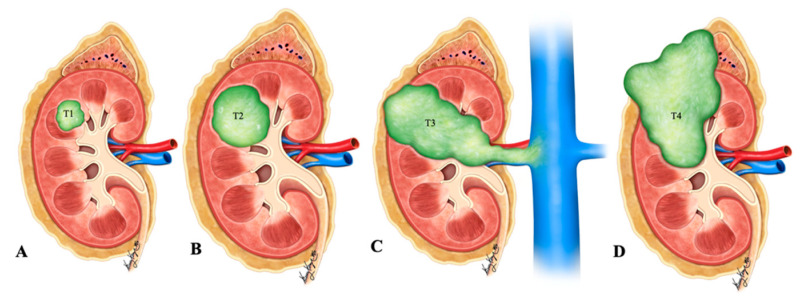
Illustration of renal neuroendocrine carcinoma. (**A**) T1: tumor limited to kidney and ≤7 cm in greatest dimension; (**B**) T2: tumor limited to kidney and >7 cm in dimension; (**C**) T3: tumor extending into the major vein or the tissue around the kidney; (**D**) T4: tumor growth beyond the Gerota’s fascia and may be growing into the adrenal gland.

**Figure 3 cancers-14-03218-f003:**
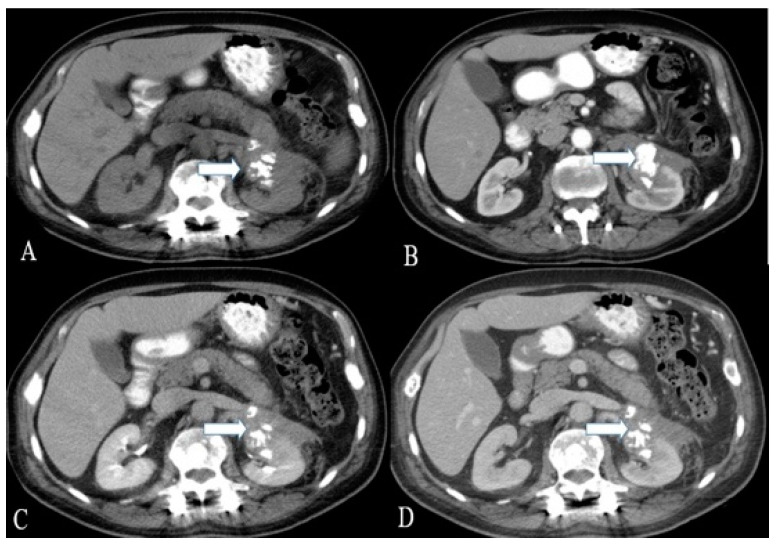
A 36-year-old female with renal NENs. CT of the abdomen before (**A**) and after oral and IV contrast administration in (**B**) arterial, (**C**) porto-venous, and (**D**) delayed phases reveal a 5.3 × 3.9 cm partially calcified mass (arrows) arising from the left kidney, extending to the left perinephric space, and abutting the tail of the pancreas. Pathology revealed small-cell carcinoma with neuroendocrine differentiation.

**Figure 4 cancers-14-03218-f004:**
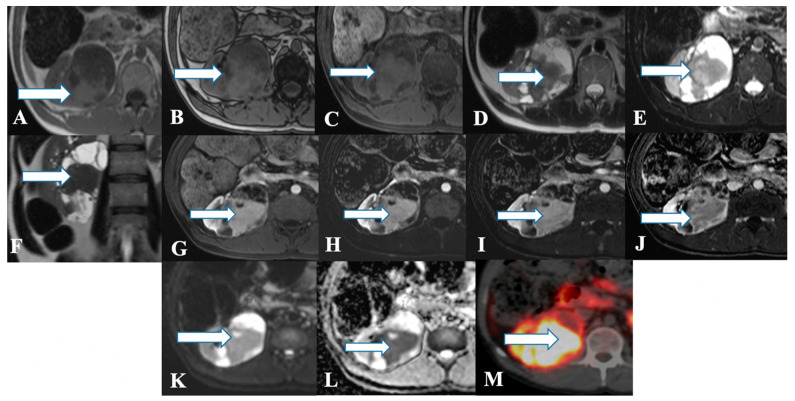
A 45-year-old female with renal NENs. MRI axial T1 (**A**) in phase, (**B**) out of phase, (**C**) 3D axial, (**D**) T2, (**E**) T2 fat saturation, (**F**) coronal T2, (**G**) post-contrast dynamic, (**H**,**I**) dynamic subtraction images, (**J**) delayed postcontrast axial, (**K**) DWI, and (**L**) ADC show a large right renal mid and lower pole mass (arrows) measuring 6.6 × 7.1 cm showing central enhancing solid component with restricted diffusion and peripheral cystic component. Pathology revealed a neuroendocrine tumor. (**M**) Axial fused PET/CT images using Gallium 68 Dotatate show heterogeneously avid DOTATATE uptake of the mass (arrow), in keeping with biopsy-confirmed right kidney neuroendocrine carcinoma.

**Figure 5 cancers-14-03218-f005:**
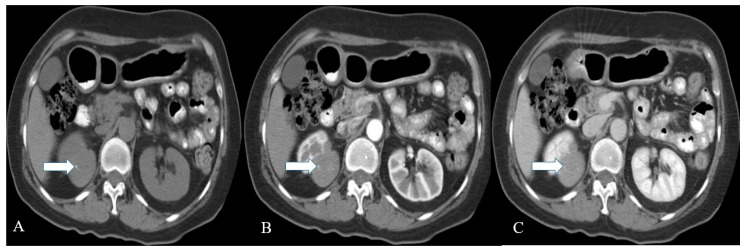
A 57-year-old female with renal NENs. CT scan of the abdomen before (**A**) and after oral and IV contrast administration in arterial (**B**) and porto-venous (**C**) phases reveal a right renal upper polar well-defined mass (arrow) with flecks of calcifications, measuring 5 × 4 cm encroaching on the right perinephric space without appreciable involvement of the right renal hilum. Pathology revealed well-differentiated neuroendocrine carcinoma (atypical carcinoid).

**Figure 6 cancers-14-03218-f006:**
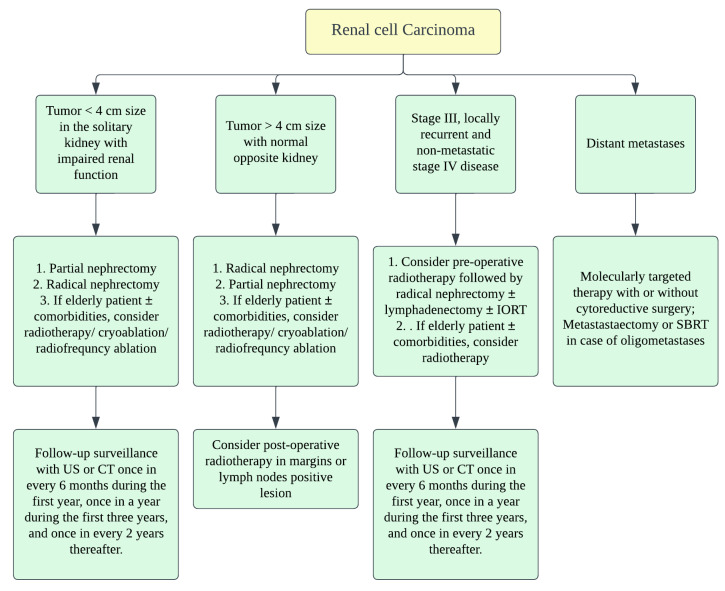
Treatment algorithm of renal malignancies.

**Figure 7 cancers-14-03218-f007:**
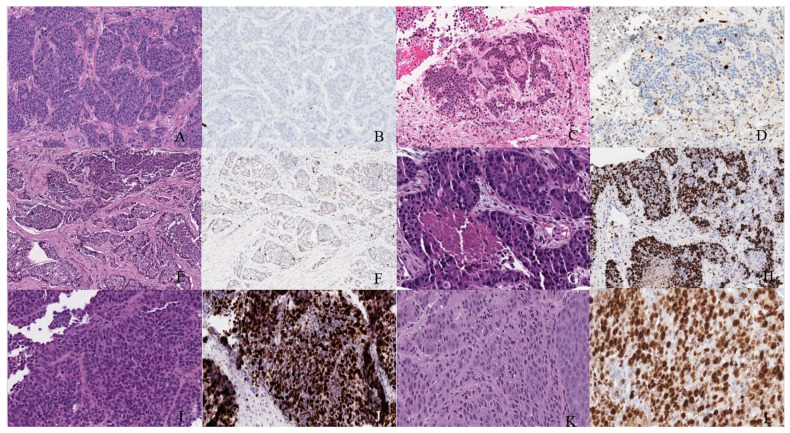
Well-differentiated NET (neuroendocrine tumor). (**A**) Grade 1 shows packets of neuroendocrine cells separated by fibrovascular tissue. (**B**) Ki67 immunohistochemical stain shows a proliferation rate of 1%. Grade 2 well-differentiated NET showing (**C**) trabecular architecture. (**D**) Ki67 immunohistochemical stain shows a proliferation rate of 15%. Grade 3 well-differentiated NET with a (**E**) solid and nested pattern. (**F**) Ki67 immunohistochemical stain shows a proliferation rate of 30%. Poorly differentiated neuroendocrine carcinoma. (**G**) The tumor shows solid nests of poorly differentiated epithelioid cells with elevated nuclear size, pleomorphism, and dense chromatin. Notice there is necrosis in the center of the image. (**H**) Ki67 immunohistochemical stain shows a proliferation rate of 80%. Small cell neuroendocrine carcinoma (**I**) sheets of small to medium, round/oval, blue cells with minimal cytoplasm. The chromatin is finely dispersed. (**J**) Nuclei demonstrate molding. Ki67 shows a proliferation index of 80%. Large cell neuroendocrine carcinoma (**K**) with organoid architecture. Large cells (~3× size of small-cell carcinoma) are present with abundant cytoplasm, variably coarse chromatin, nuclear pleomorphism, and prominent nucleoli. (**L**) Ki67 immunohistochemical stain shows a proliferation rate of 90%.

**Figure 8 cancers-14-03218-f008:**
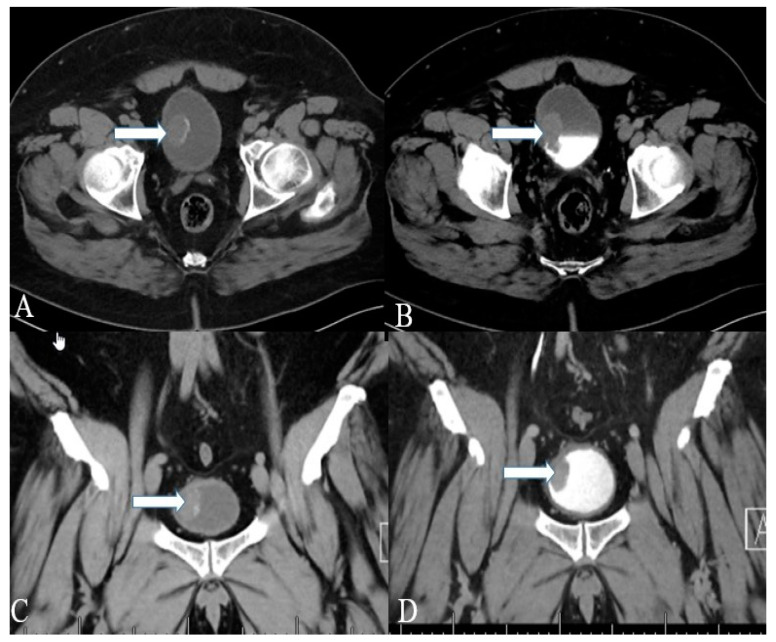
A 40-year-old female with urinary bladder NENs. Axial CT (**A**) pre-contrast and (**B**) delayed post-contrast images with coronal reconstruction (**C**) pre-contrast and (**D**) post-contrast reveal a 4.8 × 2.5 cm mass (arrows) at the proper aspect of the urinary bladder, showing faint peripheral calcification. Pathology revealed a neuroendocrine tumor of the urinary bladder.

**Figure 9 cancers-14-03218-f009:**
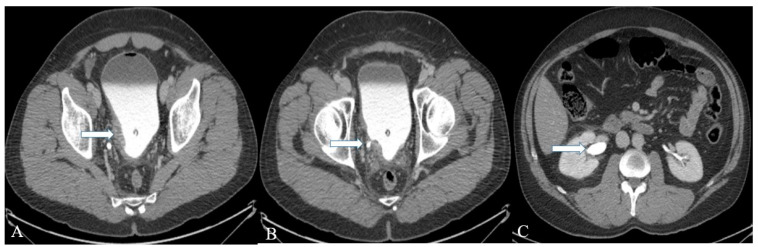
A 32-year-old female with urinary bladder NENs. Axial CT post-contrast (**A**) shows soft tissue thickening (arrow) mounting to mass formation involving the right lateral and posterior urinary bladder wall. (**B**,**C**) It is seen narrowing (arrow in (**B**)) the right ureterovesical junction with secondary mild right hydroureteronephrosis (arrow in (**C**)). This mass seems to infiltrate the perivesical fat. A catheter balloon is seen within the bladder lumen. Pathology of the group revealed urothelial carcinoma with neuroendocrine features and focal glandular differentiation, high grade.

**Figure 10 cancers-14-03218-f010:**
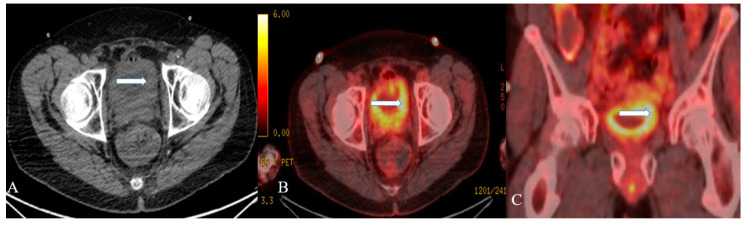
A 48-year-old female with urinary bladder NENs. Axial CT without contrast (**A**), axial (**B**), and coronal (**C**) reconstruction fused PET/CT images show thickened urinary bladder wall (arrows), which is most evident on the left side. The maximum thickness measures 2.4 cm and has an SUV of 8.2. Pathology revealed small-cell neuroendocrine carcinoma of the urinary bladder.

**Figure 11 cancers-14-03218-f011:**
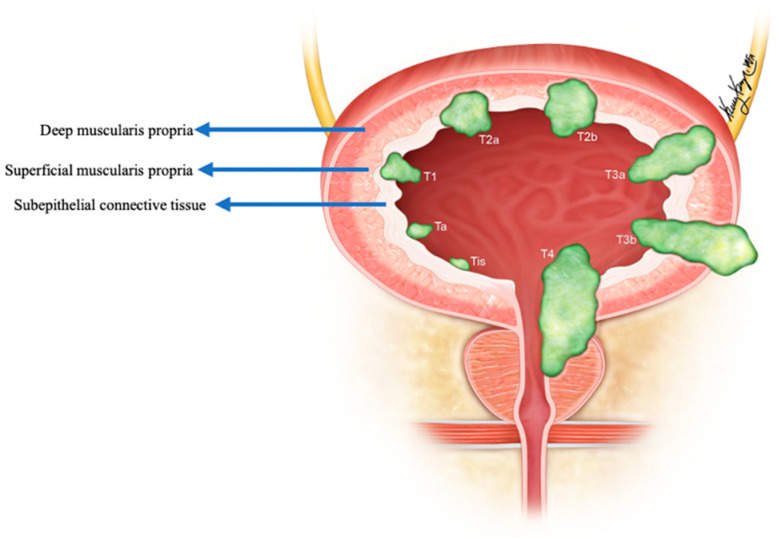
Staging of urinary bladder neuroendocrine carcinoma. Ta: non-invasive carcinoma; Tis: carcinoma in situ (Flat-tumor); T1: tumor invasion into subepithelial connective tissue; T2a: tumor invasion into superficial muscularis propria; T2b: umor invasion into deep muscularis propria; T3a: microscopic invasion of peri-vesical tissue; T3b: macroscopic invasion of peri-vesical tissue (extravesical mass); T4: tumor invasion of adjacent structures, pelvic or abdominal wall.

**Figure 12 cancers-14-03218-f012:**
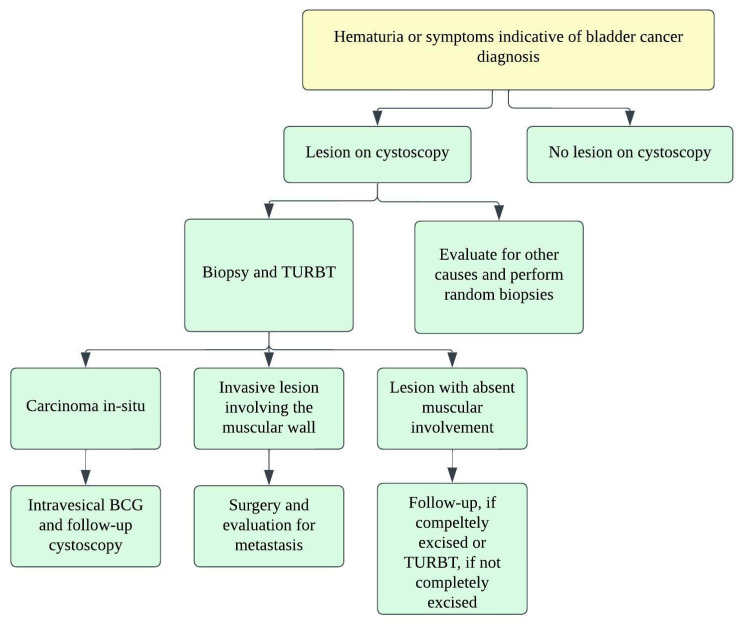
Treatment algorithm for urinary bladder carcinoma.

**Figure 13 cancers-14-03218-f013:**
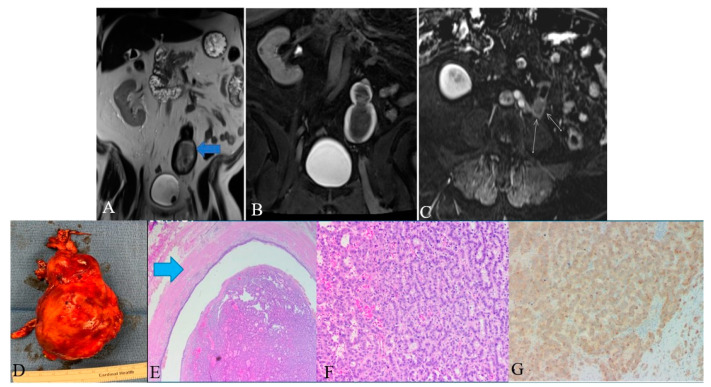
A 48-year-old female with ureteral NENs. (**A**) Coronal T2 nonfat sat image shows a large expansible hematoma in the ureteric stump (arrows). (**B**) Coronal T1 postcontrast delayed fat sat image: Shows a large expansible hematoma with intrinsic high T1 signal. (**C**) Axial T1 postcontrast delayed fat sat subtraction image: a small enhancing nodule (arrows) along the superior aspect of the stump. (**D**) Left lower ureter mass consists of a segment of the ureter with an attached firm, tan-brown, irregular, mass, measuring 11.5 × 7.0 × 6.0 cm. (**E**) E&H staining: 20× shows polypoid tumor bulging into the ureter lumen. Normal urothelial mucosa and muscle layer can be seen on the top half (arrow). (**F**) H&E straining: 200× high-power view shows tumor cells arranged in a trabecular and tubular pattern. (**G**) Synaptophysin staining: 200× shows tumor cells expressing synaptophysin (brown staining), which is a marker of neuroendocrine differentiation.

**Figure 14 cancers-14-03218-f014:**
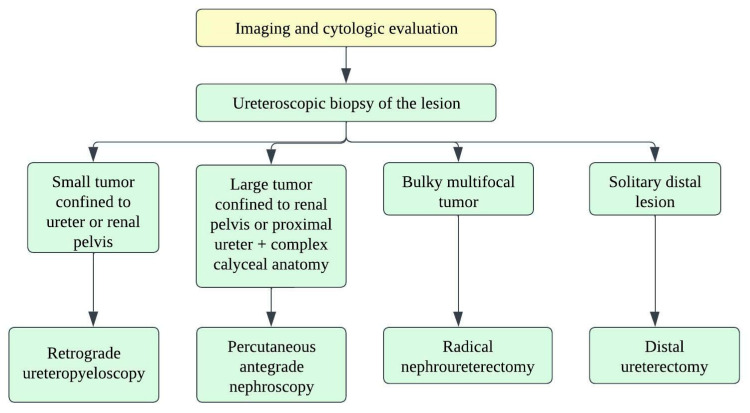
Treatment algorithm for upper urinary tract carcinoma.

**Figure 15 cancers-14-03218-f015:**
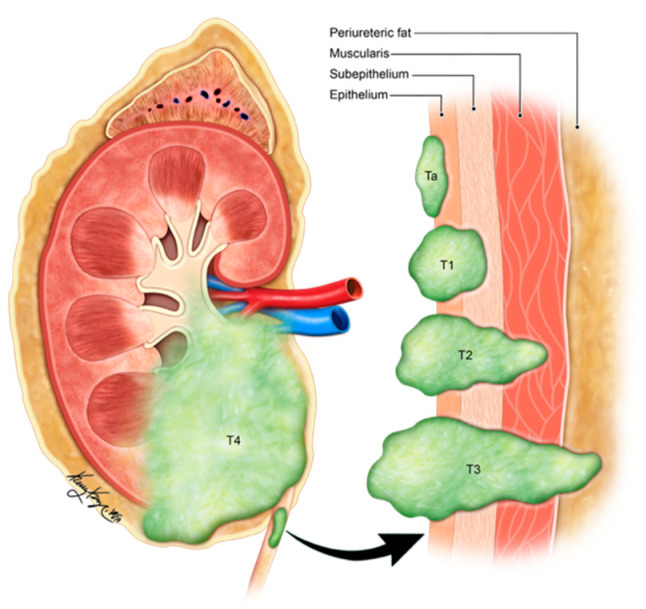
Illustration demonstrating the staging of ureteral carcinoma. Ta: non-invasive papillary tumor; T1: tumor invasion into sub-epithelial connective tissue through lamina propria; T2: tumor invasion into muscularis propria; T3: tumor invasion into periureteric fat beyond muscularis propria; T4: tumor invades adjacent organs or through the kidney into the perinephric fat.

**Figure 16 cancers-14-03218-f016:**
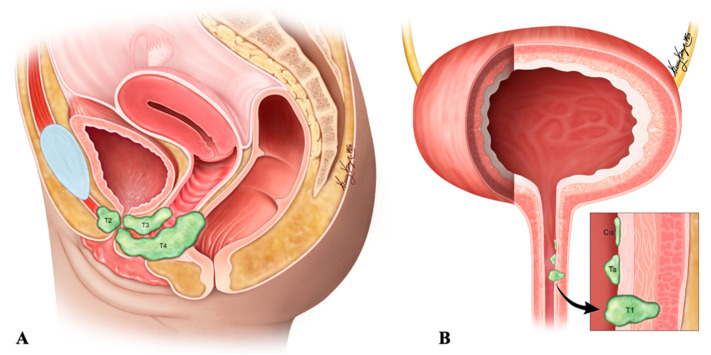
Staging of urethral neuroendocrine carcinoma. (**A**) Sagittal view of the tumor spread (**B**) Urethral site specific tumor spread. Ta: non-invasive papillary, polypoid, or verrucous carcinoma; Cis: carcinoma in situ; T1: tumor invasion into subepithelial connective tissue; T2: tumor invasion into periurethral muscles; T3: tumor invasion into anterior vagina or bladder neck; T4: tumor invasion into adjacent organs such as the bladder and rectum.

**Table 1 cancers-14-03218-t001:** Morphological characteristics of renal NENs [26,27,28,29,30].

	Renal Carcinoids	Renal Small-Cell NEC	Renal Large-Cell NEC	Renal Paraganglioma
Cellular arrangements	Trabecular/gyriform, glandular, insular, solid or mixed patterns	Sheets, nests, trabeculae		Solid nesting growth pattern
Cellular characteristics	Round or polygonal cells with granular cytoplasm.	Poorly differentiated small round to fusiform cells; scanty stroma and cytoplasm	Large cells with abundant cytoplasm	Round to oval cells (Zellballen); sometimes spindle-shaped elongated cells; abundant granular cytoplasm;
Nuclei	Round nucleus with finely stippled ribbon-like chromatin; inconspicuous nucleoli	Hyperchromatic nuclei; Nuclear molding; stippled/dispersed chromatin; inconspicuous nucleoli	Pleomorphic nuclei with vesicular chromatin; Prominent nucleoli	Variant nuclei with regular uniform chromatin or hyperchromasia
Additional features	Calcifications (25% of cases); Absence of frequent mitosis; Low Ki-67 proliferation index	Extensive tumor necrosis; perivascular DNA deposition (Azzopardi phenomenon); Brisk mitoses; Calcifications can be seen.	High mitotic rate (>10/10 per HPF); Large necrotic areas	Highly vascular intervening stroma; Low mitotic index and necrosis
Immunohistochemistry	Positive for cytokeratin, chromogranin, synaptophysin, gremileus and neuron-specific enolase	Dot-like cytokeratin staining in the cytoplasm; Variable positivity for chromogranin A, synaptophysin, CD56 and NSE; A few cases reported positive staining for TTF-1 [17,31,32]	Diffusely and strongly positive for synaptophysin	Positive for *INSM1*, synaptophysin and chromogranin; Negative for keratins; Positive tyrosine hydroxylase and nuclear *GATS3*; scattered S100-sustentacular cells

**Table 2 cancers-14-03218-t002:** Imaging features of solid renal masses [48].

Renal Lesion	CT	MRI
Clear Cell carcinoma	Heterogeneous mass due to necrotic, cystic and hemorrhagic areas; Strong contrast enhancement in corticomedullary and contrast wash out during nephrographic phases	Hyper vascular lesion; Hyperintense on T2WI and hypo- to isointense on T1WI; Heterogeneous avid enhancement than the rest of RCC types on contrast administration; Microscopic fat in 60% of cases; CSI: >25% signal loss on opposed phase relative to in-phase imaging due to fat content; Tumor pseudo capsule: hypointense rim on T1WI and T2WI.
Chromophobe carcinoma	Homogeneous to heterogeneous mass	Well circumscribed homogeneous tumors; Iso- to hypointense lesion son T2WI; The lesion enhances greater than papillary but lesser than clear cell renal carcinoma; Central stellate scar in 30–40% of cases; Spoke-wheel enhancement can be observed; Segmental enhancement inversion may be noticed; Calcifications in 38% of cases.
Papillary carcinoma	Tumors < 3 cm: homogeneous; Tumors ≥ 4 cm: Heterogeneous due to necrosis; Subtle contrast enhancement than ccRCC; Absent enhancement can be observed in 25% of patients;	Well-circumscribed homogeneous mass; Usually <3 cm; Mass: hypointense on T2WI which enhance progressively with contrast administration; CSI: signal loss on in-phase relative to opposed phase imaging due to hemosiderin deposition; Fibrous capsule: hypointense on T1WI & T2WI;
Renal NET	Heterogeneous solid tumor with cystic component as well; Minimal enhancement on contrast administration; **Octreotide scintigraphy:** High affinity for somatostatin in 87% of patients.	Heterogeneous signal intensity on T1 and T2WI with areas of high signal intensity on T1WI due to hemorrhage; The mass enhances with contrast administration

**Table 3 cancers-14-03218-t003:** AJCC staging of renal malignancies.

Stage	TNM Category	Description
I	T1 N0 M0	T1–Tumor limited to kidney & ≤7 cm in greatest dimension N0–No lymph node involvement M0–No distant metastasis
II	T2 N0 M0	T2–Tumor limited to kidney & >7 cm in greatest dimension
III	T3 N0 M0	T3–Tumor extending into the major vein or into the tissue around kidney; but not growing into adrenal gland or beyond Gerota’s fascia
	T1-3 N1 M0	N1–Tumor spread to regional lymph nodes
IV	T4 Any N M0	T4–Tumor beyond Gerota’s fascia and may be growing into adrenal gland on the top of kidney
	Any T Any N M1	M1–Distant lymph nodes and/or other organs.

**Table 4 cancers-14-03218-t004:** AJCC staging of paragangliomas [30].

Stage	TNM Category	Description
I	T1 N0 M0	T1–Pheochromocytoma < 5 cm in greatest dimension; No extra-adrenal invasion; N0–no lymph node metastases; M0–No distant metastases
II	T2 N0 M0	T2–Pheochromocytoma ≥ 5 cm, sympathetic paraganglioma of any size; no extra-adrenal invasion
III	T1 N1 M0	N1–Lymph node metastases
III	T2 N1 M0	
III	T3 N0 M0	T3–Tumor of any size with surround tissue invasion (liver, spleen, pancreas and kidneys)
III	T3 N1 M0	
IV	Any T Any N M1	M1–distant metastases; M1a–Distant metastases to only bone; M1b–Distant metastases to only lymph nodes/liver/lung; M1c–Distant metastases to bone plus multiple other sites

**Table 5 cancers-14-03218-t005:** Histological characteristics of urinary bladder NENs [64].

Histological Features	Well-Differentiated NET	Small-Cell NEC	Large-Cell NEC	Paraganglioma
Cellular arrangements	Anastomosing cords; Glandular; Cribriform structures	Diffuse sheets; Nests	Sheets; Solid nests; Trabeculae; Rosettes	Nests; Diffuse growth; Pseudo rosettes
Cellular characteristics	Intermediate cuboidal/columnar monomorphic cells with moderate to abundant cytoplasm and eosinophilic granules	Small/intermediate fusiform cells with scant cytoplasm	Large polygonal cells with abundant cytoplasm	Large polygonal cells with moderate cytoplasm
Nuclei	Small round to oval nuclei with finely stippled chromatin and inconspicuous nucleolus	Small round to oval nuclei with finely granular chromatin, molding and crush artifact; Inconspicuous salt and pepper nucleolus	Large oval nuclei with coarse, granular, vesicular chromatin and prominent nucleolus	Medium round to oval nuclei with smudged, hyperchromatic chromatin and prominent nucleolus
Additional findings	Infrequent mitotic activity and absent necrosis	High mitotic activity and foci of necrosis; Lymphovascular invasion	Very high mitotic activity and large areas of necrosis	Rare mitotic activity and necrosis

**Table 6 cancers-14-03218-t006:** AJCC staging of urinary bladder malignancy—8th edition [100,101].

Stage	TNM Category	Description
0a	Ta N0 M0	Non-invasive papillary carcinoma; No lymph node involvement; No distant metastasis
0is	Tis N0 M0	Carcinoma in-situ (“Flat-tumor”)
I	T1 N0 M0	Tumor invasion into subepithelial connective tissue (lamina propria)
II	T2a N0 M0	Tumor invasion into superficial muscularis propria (inner half of detrusor muscle)
	T2b N0 M0	Tumor invasion into deep muscularis propria (outer half of detrusor muscle)
IIIA	T3a N0 M0	Microscopic invasion of peri-vesical tissue
	T3b N0 M0	Macroscopic invasion of peri-vesical tissue (extravesical mass)
	T4a N0 M0	Tumor invades any of: prostatic stroma, seminal vesicles, uterus, vagina
	T1-4a N1 M0	Single regional lymph node involvement: peri-vesical, obturator, internal and external iliac, or sacral lymph nodes
IIIB	T1-4a N2 or N3 M0	N2–Multiple regional lymph node involvement; N3–Common iliac lymph node involvement
IVA	T4b Any N M0	T4b–Tumor invasion into pelvic or abdominal wall
	Any T Any N M1a	M1a–Distant metastases to lymph nodes beyond common iliac arteries
IVB	Any T Any N M1b	M1b–Distant metastases to sites such as bones, liver or lungs

**Table 7 cancers-14-03218-t007:** Morphological features of small-cell ureteral NEC.

Features	Small-Cell NEC
Gross	Well-defined firm-greyish mass protruding into the ureteral lumen; hemorrhagic areas
Histology	
Cellular arrangement	Solid sheets; rosette; nests
Cellular characteristics	Small to medium sized cells with scant cytoplasm and granular chromatin
Additional features	Frequent necrotic areas, mitosis and vascular invasion
Immunohistochemistry	
Neuroendocrine stains	Chromogranin A, synaptophysin, CD56, neuron-specific enolase
Epithelial stains	Cytokeratin-7, epithelial membrane antigen, and pan-cytokeratin
Differential stains	Uroplakin III-negative (positive in umbrella cells of urothelium and transitional cell carcinoma)

**Table 8 cancers-14-03218-t008:** Differential diagnosis of large-cell ureteral NEC [74,109].

Differential Diagnosis	Distinguishing Feature
Carcinoid tumors	Small cells with low-grade nuclear atypia; low mitotic activity (usually <2/HPF); low Ki-67 index
Small-cell carcinoma	Small cells (usually less than the diameter of the three small lymphocytes) with scant cytoplasm; fine granular chromatin; absent or inconspicuous nuclei; high miotic activity (≥11/HPF with a median of 80/HPF); frequent large areas of necrosis
High-grade urothelial carcinoma	Poorly differentiated cells with centrally located nuclei and thick, rough nuclear membranes; Identifiable nucleoli; Irregular chromatin; Positive immunohistochemistry for uroplakin (57–81% of cases) and negative or neuroendocrine markers;
Primary or metastatic adenocarcinoma	Diffuse glandular morphology; negative neuroendocrine markers on immunohistochemical analyses

**Table 9 cancers-14-03218-t009:** TNM staging of ureteral malignancies [123].

Stage	TNM Staging	Description
0	Ta N0 M0	Non-invasive papillary tumor
	Tis N0 M0	Carcinoma in-situ
I	T1 N0 M0	Tumor invasion into sb-epithelial connective tissue through lamina propria
II	T2 N0 M0	Tumor invasion into muscularis propria
III	T3 N0 M0	Tumor invasion into periureteric fat beyond muscularis propria
	T4 N0 M0	Tumor invades adjacent organs or through the kidney into the perinephric fat
IV	T4 Any N Any M	N1-metastasis in a single lymph node ≤ 2 cm in greatest dimension N2-Metastasis in a single lymph node > 2 cm; or multiple lymph node involvement N3-Metastasis in a lymph node, more than 5 cm in greatest dimension M1-Distant metastasis
	Any T N1-3 Any M	

**Table 10 cancers-14-03218-t010:** Differential diagnosis of small-cell urethral NEC [124,125,128].

Differential Diagnosis	Characteristic of Urethral NEN That Helps in Distinguishment
Lymphoma	Positive immunohistochemistry with LCA antigen
Transitional cell carcinoma	Positive p63 and CD44v6 (80% specific)
Carcinoid tumor	Abundant eosinophilic cytoplasm and regular nuclei
Primary neuroectodermal tumor	Larger cell bodies with spread out cytoplasm in dendritic processes
Metastatic small-cell carcinoma	Positive TTF-1 and presence of primary site on chest CT
Merkel cell carcinoma	Penile or scrotal skin involvement and presence of punctate paranuclear cytokeratin staining; positive CK20

**Table 11 cancers-14-03218-t011:** AJCC staging of urethral malignancies [129].

Stage	TNM Category	Description	
0a	Ta N0 M0	Ta-Non-invasive papillary, polypoid, or verrucous carcinoma; N0-No lymph node involvement; M0-No distant metastases	Proximal: Partial or complete urethrectomy Distal: Urethra-sparing surgery/urethrectomy/radiotherapy
0is	Tis N0 M0	Tis-Carcinoma in-situ	Proximal: Partial or complete urethrectomy Distal: Urethra-sparing surgery/urethrectomy/radiotherapy
I	T1 N0 M0	T1-Tumor invasion into subepithelial connective tissue	Proximal: Partial or complete urethrectomy Distal: Urethra-sparing surgery/urethrectomy/radiotherapy
II	T2 N0 M0	T2-Tumor invasion into corpus spongiosum/prostate or peri-urethral muscles	Proximal: Partial or complete urethrectomy ± neoadjuvant chemotherapy Distal: Urethra-sparing surgery/urethrectomy/radiotherapy
III	T1 N1 M0	N1-Single lymph node involvement ≤ 2 cm in greatest dimension	Inductive chemotherapy + consolidative surgery; chemoradiotherapy
	T2 N1 M0		Inductive chemotherapy + consolidative surgery; chemoradiotherapy
	T3 N0 M0	T3-Tumor invasion into corpus cavernosum, beyond prostate capsule, anterior vagina and bladder neck	Neoadjuvant chemotherapy + surgery; Surgery + adjuvant radiotherapy
	T3 N1 M0		Inductive chemotherapy + consolidative surgery; chemoradiotherapy
IV	T4 N0 M0	T4-Tumor invasion into adjacent organs (e.g., Bladder)	Neoadjuvant chemotherapy + surgery; Surgery + adjuvant radiotherapy
	T4 N1 M0		Inductive chemotherapy + consolidative surgery; chemoradiotherapy
	Any T N2 M0	N2-Single lymph node > 2 cm but ≤5 cm or multiple lymph node involvement (≤5 cm)	Inductive chemotherapy + consolidative surgery; chemoradiotherapy
	Any T Any N M1	M1-Distant metastases	Systemic therapy

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
