# Peer review of "Neuroendocrine Neoplasms of the Female Genitourinary Tract: A Comprehensive Overview"

_cancers, 2022, doi:10.3390/cancers14133218_

Round 1

Reviewer 1 Report

Thank you for a well-written review. 

Only few comments

Line 42: I would erase pheochromocytoma - as there are no renal phenochromocytomas. 

Libe 174: The line "Renal NEN exhibits improved survival rates than poorly differentiated renal NEN 174 (3).". Maybe that sentence needs a rephrasing?

3. Page 12, line 329. Survival rates 8-40%? Is that after one year, five years? Please add the time where these survival rates are seen. 

Author Response

  1. Line 42: I would erase pheochromocytoma - as there are no renal pheochromocytomas. 

Response: Thank you. We acknowledged and erased the term pheochromocytoma in the line. 42.

2016 WHO classification of renal tumors classified renal neuroendocrine neoplasms into well-differentiated NET, small-cell NEC, and large-cell NEC (Table 1) [4].

  1. Libe 174: The line "Renal NEN exhibits improved survival rates than poorly differentiated renal NEN 174 (3).". Maybe that sentence needs a rephrasing?

Response: Thank you. We rephrased the sentence.

Renal NEN has higher survival rates than poorly differentiated renal NEN [3].

  1. Page 12, line 329. Survival rates 8-40%? Is that after one year, five years? Please add the time where these survival rates are seen. 

Response: Thank you. We included the time for the survival rates.

The stage is the most important prognostic factor in small-cell NEC, with 5-year survival rates ranging from 8-16% in low-to-high stage disease (Figure 11) [66,93,99].

Reviewer 2 Report

The authors have compiled a comprehensive review using information from 120 citations. The manuscript reads well and is structured clearly. 

A couple minor recommendations for the authors to enhance the quality of the manuscript:

- The authors recommend AJCC based staging for the tumor types. It would be beneficial to also discuss FIGO staging for site-specific tumors.  

- A few treatment algorithm figures illustrate to observe in specific scenarios. Please explain/elaborate case follow-up timeline in the case of "observe". 

A very minor suggestion for the treatment algorithm flowchart (not a mandatory change required for acceptance) : look into flowchart creation specific tooling for better styling/decision points etc. An example online tool is Lucidchart. 

Author Response

  1. The authors recommend AJCC-based staging for the tumor types. It would be beneficial to also discuss FIGO staging for site-specific tumors. 

Response: Thank you for the comment. We included the updated AJCC staging of the genitourinary tumors.

  1. A few treatment algorithm figures illustrate to observe in specific scenarios. Please explain/elaborate case follow-up timeline in the case of "observe".

Response: Thank you. We included the timeline for the surveillance in the revised manuscript.

Follow-up surveillance with the US or CT once every 6 months during the first year, once in a year during the first three years, and once every 2 years thereafter.

  1. A very minor suggestion for the treatment algorithm flowchart (not a mandatory change required for acceptance: look into flowchart creation specific tooling for better styling/decision points etc. An example online tool is Lucidchart. 

Response: Thank you. We utilized the Lucidchart tool to construct flowcharts in the revised manuscript.